biochemistry/chemical biology/environmental chemistry

Alkane biodegradation, Functional Fe₃O₄ nanoparticles, Immobilization cell, *Rhodococcus*, Reusability

**Authors for correspondence:**
Xiaolei Ma
e-mail: huandaoyu@126.com
Baolong Xie
e-mail: xiebaolong@tju.edu.cn

This article has been edited by the Royal Society of Chemistry, including the commissioning, peer review process and editorial aspects up to the point of acceptance.

# Degradation of *Rhodococcus erythropolis* SY095 modified with functional magnetic Fe₃O₄ nanoparticles

Xiaolei Ma[1], Duomo Duan[2], Xunliang Wang[1], Junrui Cao[1], Jinquan Qiu[1] and Baolong Xie[1]

[1]Microbiology and Biotechnology Research Laboratory, The Institute of Seawater Desalination and Multi-Purpose Utilization, Ministry of Natural Resources of the People's Republic of China (MNR), Tianjin 300192, People's Republic of China
[2]Tianjin Rehabilitation Center, The PLA Joint Logistic Support Force, Tianjin, 300191, People's Republic of China

(iD) XM, 0000-0003-3626-2734

Alkali-surfactant-polymer flooding technology is widely employed to extract crude oil to enhance its production. The bacterial strain *Rhodococcus erythropolis* SY095 has shown high degradation activity of alkane of crude oil. In the past, many treatment strategies have been implemented to reduce oil concentration in wastewater. Previous studies mainly focused on the extracellular products of *Erythrococcus* rather than its degradation properties. In the current study, we designed an immobilization method to modify the surface of *R. erythropolis* SY095 with functional Fe₃O₄ nanoparticles (NPs) for biodegradation of crude oil and separation of the immobilized bacteria after degradation. We characterize the synthesized NPs through various methods, including scanning electron microscope energy-dispersive spectrometer, Fourier transform infrared spectroscopy, X-ray diffraction (XRD) and a vibrating sample magnetometer. We found that the size of the synthesized NPs was approximately 100 nm. Our results showed that *R. erythropolis* SY095 was successfully coated with functional magnetic NPs (MNPs) that could be easily separated from the solution via the application of an external magnetic field. The coated cells had a high tolerance for heavy metals. Our findings demonstrated that the immobilization of MNPs to bacterial surfaces is a promising approach for the degradation of crude oil.

## 1. Introduction

Crude oil is a complex mixture containing many thousands of different hydrocarbon compounds, which can be divided into

four classes (saturated hydrocarbons, aromatic hydrocarbons, asphaltene and non-hydrocarbons). The biodegradation of crude oil by natural populations of microorganisms was reported over a century ago. Along with the rapid industrial development, plenty of wastewaters containing heavy metals (e.g. lead, mercury, copper, cadmium and chromium) are produced, especially in the fields of electroplating, leather-making, textile dyeing and printing, and battery production [1–4]. Unlike organic pollutants, heavy metals cannot be decomposed. As a result, if the wastewaters are unable to be properly treated, heavy metals will always exist in the environment or accumulate in organisms through the food cycle [5], resulting in serious damage to environmental safety and human health because of their high toxicity and carcinogenicity.

Alkali-surfactant-polymer flooding technology is extensively used in the extraction of crude oil to enhance its production [6,7]. As crude oil contains natural surfactants (asphaltenes, resins), added chemical surfactants and alkali salts, this well-recognized technique produces oil-in-water (o/w) type emulsions [8–11]. In addition, spillage from oil tankers, oil ships, tanker collisions and oil well outbursts produce an exorbitant amount of oil sewage [12,13]. This oil waste directly affects the environment; thus, its removal before its discharge into the environment is essential. In recent years, oil pollution caused by oil sewage discharge has become one of the most serious environmental concerns [14,15]. Oil-induced pollution can cause irreversible damage to the marine environment, such as the death of aquatic organisms, and port and coastal pollution [6].

In the past, many treatment strategies including air flotation [16,17], electric coalescence [18,19], reverse osmosis [20], filtration/membrane separation, coagulation sedimentation [21–23] and bioremediation [24,25] have been implemented to reduce oil concentration in wastewater. More research is being done to explore the potential of bacteria in the degradation of crude oil. It has been reported that members of the genus *Rhodococcus* can degrade long-chain alkanes by separating them from soil or activated sludge and converting them into carbon dioxide and water [26]. As a versatile genus, *Rhodococcus* can be commonly detected in soil, air, water and animals. Moreover, it is known for its potential for degrading oil, sludge and complex hydrocarbons [27].

*Rhodococcus oryzae* has been previously reported as an effective antidote for a variety of exogenous compounds, such as phenol, cresol, monochlorophenol, dichlorophenol, triclosan, catechol and toluene [28]. However, free bacterial cells lose their activity and are difficult to recover from the polluted application sites when used for the reduction of harmful pollutants; these factors act as major obstacles in the development of an effective remedy for the removal of these exogenous compounds using microbes [29]. These issues can be overcome by immobilization of the bacterial cells, as immobilization increases the endurance of the bacteria to noxious compounds, improves catalytic efficiency and allows catalytic cell recycling [30]. Nanoparticles (NPs) are considered as the hotspot among various fields of science. Various methods for the synthesis of NPs include conventional methods like hydrothermal method, sol–gel method, laser-ablation, electrochemical techniques and thermal methods [31–35]. The current methods of immobilization and subsequent separation of bacterial cells by magnetic NPs (MNPs) are creative and promising [36]. Compared with other methods, this method exhibits lower diffusion limitation and space potential resistance [37,38]. In addition, *in vitro* cytotoxicity experiments have confirmed the biocompatibility of $Fe_3O_4$ NPs [39,40]. Due to the potential risk of the genetically engineered bacteria that are capable of degradation of pollutants to the ecosystem, it is important to recover these microorganisms from the application site using magnetic separation. Previous studies have reported biological desulphurization [38–42], phenol-degradation [43] and hexavalent chromium reduction [43] by bacterial cells wrapped by super-paramagnetic NPs (greater than 10 nm). However, the synthesis of MNPs with consistent shapes and proven dispersion remains a challenge. These issues are caused by the tendency of the particles to accumulate in the solution. Hence, for the dispersion of MNPs, silica is widely used for encapsulation or stabilization with a surfactant [44].

Low-dimensional nanomaterials offer dazzling prospective applications in a multitude of sectors. The surface of NPs plays a crucial role in their applications. Under many circumstances, surface modification of nanomaterials is needed to endow them with unique or better capabilities. The surface modification of inorganic one-dimensional nanomaterials by organic molecules can be easily done, decorating on the surface of organic semiconductor one-dimensional nanomaterials by organic molecules [45]. The surface modification of small organic molecules and high polymer can inhibit the aggregation of $Fe_3O_4$ and increase its surface effect. Previous studies found that the removal efficiency of pollutants in wastewater was significantly improved by introducing active groups on the surface of magnetic $Fe_3O_4$ NPs modified by organic small molecules and organic macromolecules [46–48]. In this study, small organic molecules (sodium citrate, oleic acid (OA) and aminopropyltriethoxysilane (APTES))

and organic polymers (D-dextran) were used to modify the MNPs. Sodium citrate and OA can provide a large amount of –COOH, which can change the stability of magnetic $Fe_3O_4$ NPs in water. Among them, sodium citrate-modified particles have improved hydrophilicity and dispersion, whereas OA-modified particles have better lipophilicity and are easier to bind to oil pollution contaminants. –OH (from dextran) and –$NH_2$ (from APTES) have strong complexing ability, in which –OH can make the particles have stronger metal cation adsorption function and the surface-modified particles with –$NH_2$ can combine with metal anions through electrostatic adsorption under acidic conditions. In this study, the above polymers containing functional groups (hydroxyl, carboxyl, amino) were used to coat the cell surface of SY095, and the effects of different active groups on cells were evaluated by investigating their degradation performance in oily sewage and heavy metal sewage.

Previously, we screened a hexadecane degrading strain and applied culture preservation in China General Microbiological Culture Collection Center for the strain (CGMCC no. 10724)—*Rhodococcus erythropolis* SY095, which could also use crude oil as a carbon source. In our previous studies, the effects of the surfactant on the solubility of n-hexadecane, the degradation efficiency of n-hexadecane, the growth of the bacteria and the hydrophobicity of the bacteria were studied [49]. In addition, we also studied the removal of lead and copper from sediments by biosurfactants from SY095 [50]. Previous studies mainly focused on the extracellular products of *Erythrococcus* rather than its degradation properties. Therefore, in this study, this strain will be used as a substrate for MNP coating, and the influence of MNPs with different surface groups on the biodegradation ability of SY095 will be studied, as well as whether it has a protective effect on SY095 from heavy metal toxicity.

# 2. Material and methods

## 2.1. Materials

MilliQ water was used for all experiments performed in this study. The purity of reference standards, n-Hexadecane, OA and citric acid (CA), was more than 99.5% (Sigma Aldrich, Germany). Dextran (approx. equal to 50 kD) was obtained from Aladdin (Shanghai, China). 3-APTES was procured from TCI (Tokyo, Japan). $Fe_3O_4$ NPs (100 nm, purity greater than 99.5%) were procured from Aladdin (Shanghai, China). $Fe_3O_4$ NPs (20 nm) were purchased from XFNANO Co. (Nanjing, China). The secondary recovery wastewater was obtained from Dagang Oilfield in Tianjin, China. Unless specified, all chemicals used in this research were of analytical grade.

## 2.2. Synthesis of $Fe_3O_4$@OH, hydrophilic-$Fe_3O_4$@COOH (OA), lipophilic-$Fe_3O_4$@COOH (CA) and $Fe_3O_4$@$NH_2$ NPs

A series of decoration methods were used to prepare functional MNPs by gradually changing their hydrophobicity.

For the synthesis of $Fe_3O_4$@OH NPs, 2 g $Fe_3O_4$ NPs and 2 g dextran were dispersed in 20 ml 0.075 M Na-citrate (sodium citrate), separately. After 30 min of ultrasonic dispersion for mixing the two solutions, they were stirred at 85°C for 2 h. The prepared MNPs were washed four times with ultrapure water, dried under vacuum and separated by magnetic adsorption.

For the synthesis of hydrophilic-$Fe_3O_4$@COOH NPs, we dispersed 20 g $Fe_3O_4$ NPs and 4 g CA in 20 ml ultrapure water, separately. The two solutions were dispersed ultrasonically for 30 min, mixed, and the pH of the solution was adjusted to 4.5–5 with 5 M NaOH; the reaction was carried out under ultrasonication for 30 min, and then the reaction was carried out under stirring at 95°C for 90 min. The prepared MNPs were washed four times with ultrapure water, dried under vacuum and separated by magnetic adsorption.

For the synthesis of lipophilic-$Fe_3O_4$@COOH NPs, we dispersed 2 g $Fe_3O_4$ NPs in 20 ml n-hexane and dissolved 1 ml OA in 20 ml n-hexane. After 30 min of ultrasonic dispersion, the two solutions were mixed by ultrasonication for 5 min, and the reactants were stirred at 60°C for 2 h. The prepared MNPs were washed four times with ultrapure water, dried under vacuum and separated by magnetic adsorption.

For the synthesis of $Fe_3O_4$@$NH_2$ NPs, 100 mg $Fe_3O_4$ NPs were dispersed in 20 ml absolute ethyl alcohol. After 10 min of ultrasonic dispersion, 80 ml alcohol and 1 ml ultrapure water were added to the solution. After 20 min of ultrasonication, 2 ml APTES was added dropwise for 1 min, and the solution was stirred at 60°C for 7 h to react. The prepared MNPs were washed four times with alcohol, dried in a vacuum and separated by magnetic adsorption.

The concentration of MNPs was expressed as dry weight per unit volume.

## 2.3. Characterization of the functionalized $Fe_3O_4$ NPs

We prepared KBr pellets by adding 1.5 mg of each NPs sample and 15 mg KBr for Fourier transform infrared spectroscopy (FTIR) analysis (Bruker). The spectral recording range was 450–4000 $cm^{-1}$. A vibrating sample magnetometer of Lakeside 7410, USA, was used for studying the magnetic attributions of the immobilized cells. X-ray diffraction (XRD) measurements were carried out at 23–25°C on the D/Max 2550VB X-ray diffractometer (Rigaku Company, Japan). A scanning electron microscope (SEM) was used for morphological observation of $Fe_3O_4$ NPs and functionalized $Fe_3O_4$ NPs. The size of the particles obtained via SEM was analysed through the ACCORD nano microscope software. The dynamic light scatterer Malvern Nano ZS90 was used to test the zeta potential of two kinds of magnetic particles at pH values of 3, 5, 7, 9 and 11. The required pH was adjusted by HCl (0.1 mol l$^{-1}$) and NaOH (0.1 mol l$^{-1}$), and the pH value was accurately measured by a pH acidity meter. Zeta potential test temperature was kept at 25°C, and each measurement includes at least 30 groups of parallel and automatic calculation by Dispersion Technology software version 4.2.

## 2.4. Culture conditions and immobilization

*Rhodococcus erythropolis* SY095 was cultured in 10 ml LB medium at 30°C for 24 h. Then, 1 ml seed culture was transferred to 100 ml LB medium and cultured for 12 h until it reached the medium exponential growth phase. The dry weight of the cells was approximately 3.96 g. Then, the cells were centrifuged at 1500$g$ for 10 min and washed three times with normal saline. The pellet was then resuspended in the following mineral medium (g l$^{-1}$: NaNO_3, 2.5; K_2HPO_4, 0.5; KH_2PO_4, 1.0; CaCl_2, 0.01; KCl, 0.1; MgSO_4 · 7H_2O, 0.5; FeSO_4 · 7H_2O, 0.01; yeast extract, 0.1; and 0.5 ml l$^{-1}$ trace element solution containing (g l$^{-1}$): H_3BO_3, 0.26; CuSO_4 · 5H_2O, 0.5; MnSO_4 · H_2O, 0.5; MoNa_2O_4 · 2H_2O, 0.06; ZnSO_4 · 7H_2O, 0.7.

The cells were coated with modified $Fe_3O_4$ NPs. We used an ultraviolet spectrophotometer (Eppendorf, Germany) for the measurement of the optical density of the bacterial cells at 600 nm (OD600) to monitor their growth rate. The optimum weight ratio for cell immobilization: dry cell weight (DCW): modified $Fe_3O_4$ NPs was 1 : 2. According to figure 5$a$, the DCW was calculated from the OD value reduction. When the bacterial cells were cultured for 36 h, the DCW of the bacterial cell was 22.4 g l$^{-1}$; the optical density of the bacterial cells at 600 nm (OD$_{600}$) was 0.72. The bacterial cells (from 100 ml) were resuspended in a fourfold volume of mineral medium, and then the MNPs (4.48 g) were added to the cell (DCW 2.24 g) suspension and oscillated for 6 h. After that, the cells coated with MNPs were separated from the uncoated cells by an external magnetic field. The collected magnetic cells were resuspended in a PBS buffer solution and stored at 4°C for a short time.

## 2.5. Ethanol-treated bacteria

The bacteria cells in the TBS buffer were collected by centrifugation at 4000$g$ for 10 min at 4°C, resuspended in 95% ethanol for 5 min and then centrifuged at 4000$g$ for 10 min. The bacteria were then washed three times with TBS buffer with pH = 7.0 and then resuspended in TBS buffer with pH = 7.0.

## 2.6. Capture rate analysis

Cells were suspended in the TBS buffer pH = 7, then mixed with the concentration of 0.2 mg ml$^{-1}$ $Fe_3O_4$ solution, oscillated for 12 h at room temperature, then the external magnetic field was applied to remove the bacteria cells and the rest of the magnetic particles. Ultraviolet spectrophotometer was used to detect the optical density value at the wavelength of 600 nm. The OD600 measured could reflect the concentration of bacterial liquid. The calculation formula is as follows:

$$MCE\,(100\%) = \frac{OD_{original} - OD_{post\ trapping}}{OD_{original}} \times 100,$$

$OD_{original}$: the initial OD$_{600}$ of bacteria before adding MNPs.
$OD_{post\ trapping}$: the optical OD$_{600}$ of the remaining bacteria in the solution after magnetic separation.

## 2.7. Biodegradation activity of the multiple immobilized cells (*Rhodococcus erythropolis* SY095 with modified Fe$_3$O$_4$ NPs)

The degradation capability of the free cells of *R. erythropolis* SY095 was tested by using crude oil or hexadecane as the sole carbon source in the mineral medium. The *R. erythropolis* SY095 cells coated with modified Fe$_3$O$_4$ NPs were cultivated in the mineral medium and 1% (v/v) crude oil or hexadecane as a carbon source. Each culture flask was incubated at 30°C and 160 r.p.m. for 72 h. Fe$_3$O$_4$ NPs of the same volume acted as the control group. The concentration of the residual crude oil in the medium was measured every 8 h; the residual crude oil was extracted by petroleum ether, and the OD of the extracted solution was monitored at 254 nm (OD$_{254}$) using an ultraviolet spectrophotometer (Eppendorf, Germany).

## 2.8. Analysis of the degradation products by multiple immobilized cells with modified Fe$_3$O$_4$ NPs

The sample medium with free or immobilized cells was extracted by petroleum ether, flushed with nitrogen and dissolved in n-hexane. The filtrate was filtered using a 0.22 mm membrane (American Millipore) before injection (10 μl) into the GC/MS system. GC/MS settings were as follows: HP-5 MS column using helium as carrier gas and 1.0 ml min$^{-1}$ flow rate (US, 30 m × 250 mm × 0.25 mm); injector temperature: 230°C; interface temperature and ion source temperature: 250°C; column temperature was increased from 100°C for 1 min, raised at 2°C min$^{-1}$ to 200°C, after that increased at 10°C min$^{-1}$ to 300°C (for 5 min); positron ionization mode was adopted for mass spectrometry analysis, and the electron ionization energy was 70 eV. Mass spectrometer identified the metabolites that matched the standard compounds in the database of the National Institute of Standards and Technology (NIST, USA).

## 2.9. Reusability of the immobilized cells for secondary oil recovery from wastewater

For the assessment of the reusability of the SY095 cells immobilized on decorated Fe$_3$O$_4$ NPs, a 3-day crude oil (1%, v/v) degradation cycle test was performed. We chose this time point because behavioural changes of bacterial cells are better reflected around the endpoint. At the end of every cycle, the immobilized cells were extracted by centrifugation at 500*g* for 5 min. The old medium was discarded, and the immobilized cells were cleaned three times with normal saline. The immobilized cells were added to a fresh medium for another intermittent cycle of degradation. The degradation efficiency of the immobilized cells was defined as the proportion of crude oil that was degraded.

# 3. Results and discussion

## 3.1. Properties of modified magnetic Fe$_3$O$_4$ NPs

### 3.1.1. Fourier transform infrared spectroscopy spectrum analysis

The solid samples dispersed in the KBr matrix were analysed by FTIR spectroscopy to characterize the dextran-modified NPs, OA-modified NPs, CA-decorated NPs and APTES-decorated NPs. Figure 1 shows the spectrum of the decorated Fe$_3$O$_4$ NPs. The intense band around 580 cm$^{-1}$ (figure 1*a–d*) represents the Fe-O stretching vibrance model of Fe$_3$O$_4$ [41]. The strong band at 3500 cm$^{-1}$ represents the presence of unseparated OH groups as previously reported [42]. As shown in figure 1*a*, the peaks at 1395 and 1000 cm$^{-1}$ signify the presence of –CH$_2$ and C–H, and C–O vibrations of dextran, respectively, which suggests that the surface of the Fe$_3$O$_4$ NPs was coated with dextran. In figure 1*b*, the peak at 1395 cm$^{-1}$ represents –CH$_2$ vibration of the CA coating layer, which suggests that the surface of the Fe$_3$O$_4$ NPs was wrapped by CA. In figure 1*c*, the peaks at 2919 and 2851 cm$^{-1}$ represent methylene asymmetric and symmetric vibration absorption, respectively, which is similar to the results from a previous study [51]. The peak at 1408 cm$^{-1}$ represents the C–O group of the OA coating layer. The peak at 951 cm$^{-1}$ represents the -OH out-of-plane ring bend vibrations, and the peak at 1054 cm$^{-1}$ could be because of Fe–O–C=O vibration, which suggests that the carboxyl group was attached to the

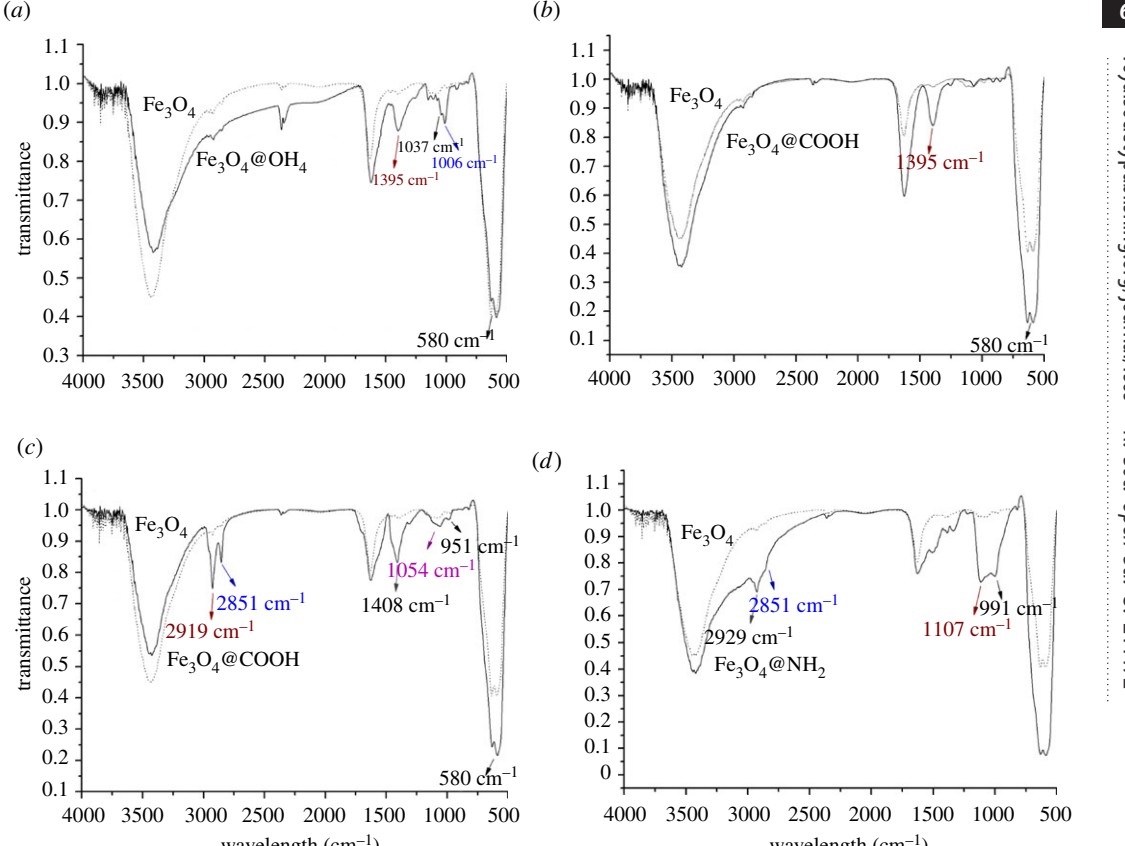

**Figure 1.** FTIR spectra of modified $Fe_3O_4$ NPs. The solid samples dispersed in the KBr matrix were analysed by FTIR spectroscopy to characterize the dextran-modified NPs, OA-modified NPs, CA-decorated NPs and APTES-decorated NPs. (a) $Fe_3O_4$ and $Fe_3O_4$@OH NPs (dextran-coated), the peaks at 1395 and 1000 cm$^{-1}$ signify the presence of $-CH_2$ and C–H, and C–O vibrations of dextran, (b) $Fe_3O_4$ and $Fe_3O_4$@COOH NPs (CA-coated), the peak at 1395 cm$^{-1}$ represents $-CH_2$ vibration of the CA coating layer, (c) $Fe_3O_4$ and $Fe_3O_4$@COOH NPs (OA-coat), the peaks at 2919 and 2851 cm$^{-1}$ represent methylene asymmetric and symmetric vibration absorption and (d) $Fe_3O_4$ and $Fe_3O_4$@NH$_2$ NPs (APTES-coated) shows that the peaks at 2929 and 2853 cm$^{-1}$ peak represent $CH_2$ and C–H vibrance absorption.

surface of MNPs through covalent bonds [52]. These results indicate that the $Fe_3O_4$ NPs were coated by OA. Figure 1$d$ shows that the peaks at 2929 and 2853 cm$^{-1}$ peak represent $CH_2$ and C–H vibrance absorption, respectively, of APTES, according to a previous report [33]. The peak at 1107 cm$^{-1}$ peak represents asymmetric vibrance absorption of Si-O-Si. The band at 991 cm$^{-1}$ was because of symmetric vibrance absorption of Si-O-Si from APTES. These results indicate that the $Fe_3O_4$ NPs were coated with amino-silane molecules by Fe-O-Si linkage.

### 3.1.2. X-ray diffraction analysis

XRD was used for the determination of the chemical constitution of the MNPs. The particle morphology was the same as that of standard $Fe_3O_4$. Figure 2$a$ shows the XRD figures of $Fe_3O_4$ NPs and decorated $Fe_3O_4$ NPs. The specific peaks at 2θ values of 30, 35.5, 43.4, 53.5, 57.2 and 62.8 show the cubic spinal mix of the magnet.

### 3.1.3. Vibrating sample magnetometer analysis

As shown in figure 2$b$, the saturation magnetizations (Ms) of bare $Fe_3O_4$, $Fe_3O_4$@OH, $Fe_3O_4$@COOH (OA-coated), $Fe_3O_4$@NH$_2$ and $Fe_3O_4$@COOH (CA-coated) were 38.20, 38.24, 40.69, 38.07 and 37.92 emu g$^{-1}$, respectively (figure 2$b$). Ms increased during OA modification, but there was no significant change during citric acid modification. The saturation magnetization of MNPs modified by OA increased slightly, which was due to the decrease of crystallinity of MNPs modified by OA. In addition, the surface modification can change the particle size of MNPs, which may be the reason for the increase

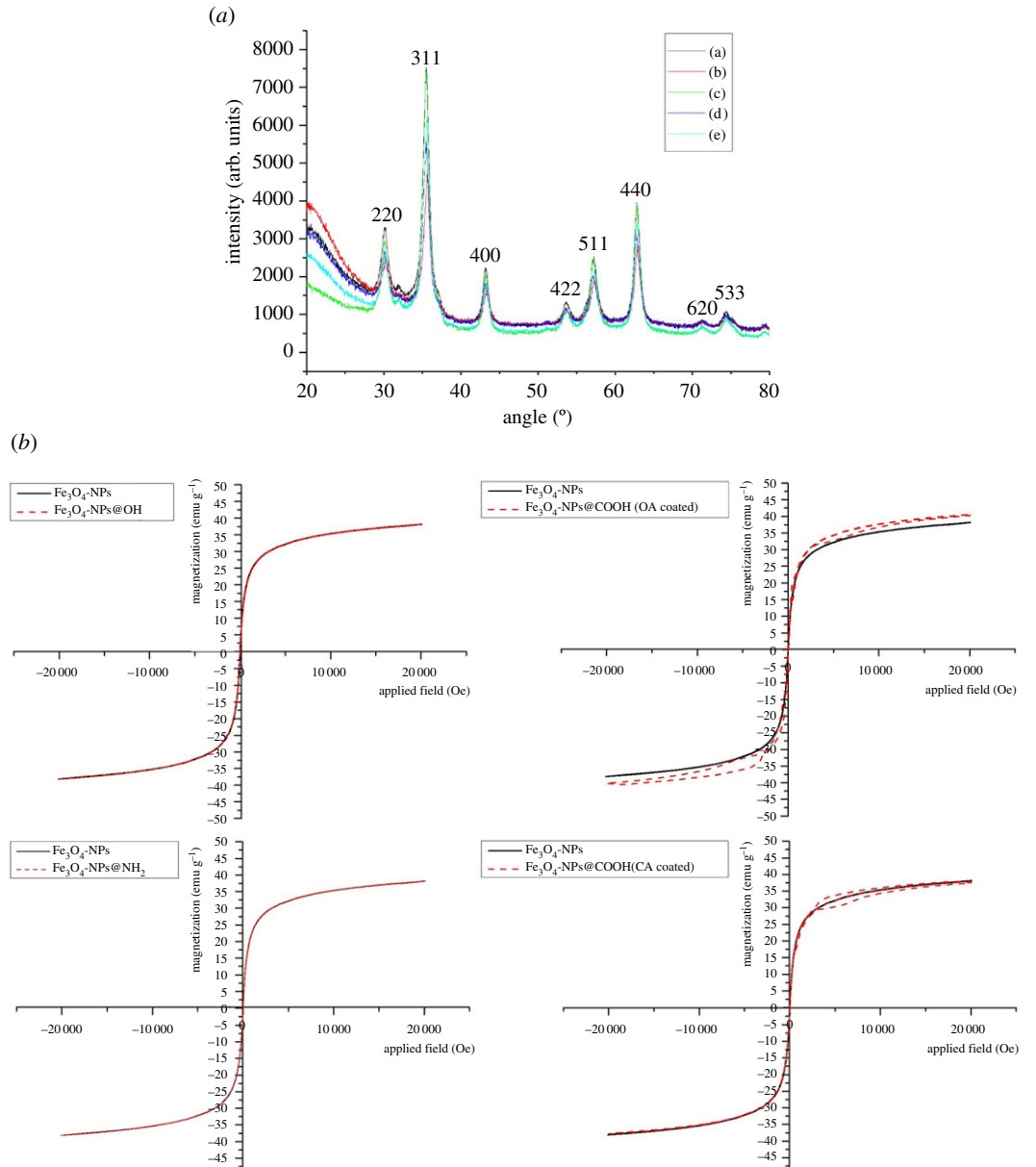

**Figure 2.** (*a*) Wide-angle XRD patterns of Fe$_3$O$_4$ NPs and modified Fe$_3$O$_4$ NPs (a) Fe$_3$O$_4$, (b) Fe$_3$O$_4$@COOH NPs (CA-coated), (c) Fe$_3$O$_4$@COOH NPs (OA-coated), (d) Fe$_3$O$_4$@OH NPs and (e) Fe$_3$O$_4$@NH$_2$ (APTES-coated) NPs. (*b*) Magnetization curves of Fe$_3$O$_4$ NPs and modified Fe$_3$O$_4$ NPs.

of saturation magnetization of MNPs after OA modification. Moreover, the magnetic hysteresis loops of Fe$_3$O$_4$@COOH (OA-coated) NPs and Fe$_3$O$_4$@COOH (CA-coated) NPs were not completely coincident, which indicated that OA and CA modification processes affected the super-paramagnetism of the Fe$_3$O$_4$ NPs. Dextran and APTES modification processes barely affected the magnetization of the NPs. Therefore, the prepared makings were majorly super-paramagnetic, and so their coercive force and residual magnetism could be ignored. The result shows that the modified Fe$_3$O$_4$ NPs can be isolated through an external magnetic field.

### 3.1.4. Morphology

We used a scanning electron microscope energy-dispersive spectrometer (SEM-EDS) for morphological characterization and elemental analysis of the Fe$_3$O$_4$ NPs and decorated Fe$_3$O$_4$ NPs (figure 3).

The microscopic morphology of Fe$_3$O$_4$ NPs was compared with the decorated Fe$_3$O$_4$ NPs. Figure 3 shows the SEM images of Fe$_3$O$_4$ NPs and decorated Fe$_3$O$_4$ NPs. The sizes of both types of NPs did not change significantly at a magnification of 180 000×. At a magnification of 30 000×, the

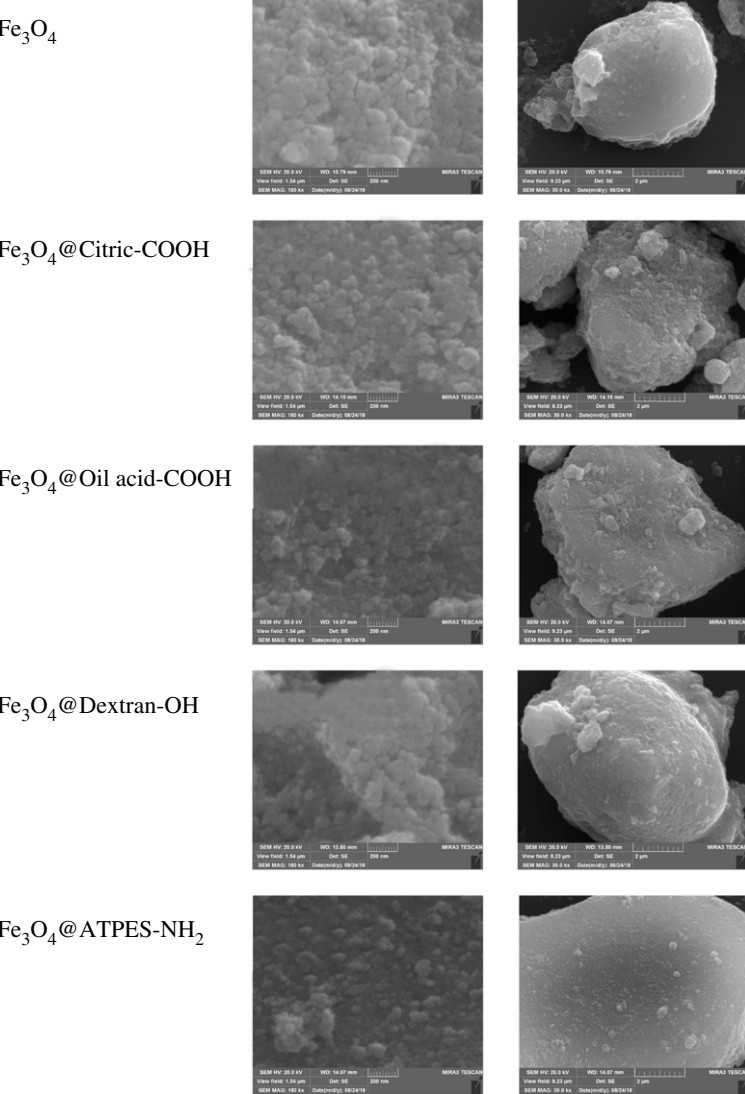

**Figure 3.** SEM images of Fe$_3$O$_4$ NPs and modified Fe$_3$O$_4$ NPs.

**Table 1.** EDS elemental analysis of Fe$_3$O$_4$ NPs and modified Fe$_3$O$_4$ NPs.

| element | Fe$_3$O$_4$ NPs (%) | Fe$_3$O$_4$@COOH (%) | Fe$_3$O$_4$@COOH(OA) (%) | Fe$_3$O$_4$@OH (%) | Fe$_3$O$_4$@NH$_2$ (%) |
| --- | --- | --- | --- | --- | --- |
| C | 23 | 41 | 31 | 29 | 28 |
| O | 45 | 58 | 50 | 50 | 51 |
| Fe | 32 | 1.3 | 19 | 20 | 22 |

agglomerate surface of Fe$_3$O$_4$ NPs was smooth, which was similar to that of dextran- and APTES-modified Fe$_3$O$_4$ NPs. Compared with the Fe$_3$O$_4$ NPs, the agglomerates of OA- and CA-modified Fe$_3$O$_4$ NPs showed rough surfaces and irregular shapes; the latter was particularly obvious. The modification process of OA and citric acid influenced the super-paramagnetism of Fe$_3$O$_4$ NPs, which is consistent with the results of the magnetization curves.

SEM-EDS was used to analyse the elements of Fe$_3$O$_4$ NPs and decorated Fe$_3$O$_4$ NPs for studying the surface modification of Fe$_3$O$_4$ NPs. Table 1 shows that the iron content on the surface of the decorated particles decreased in comparison with that of the undecorated NPs. The iron content on the surface of the CA-modified Fe$_3$O$_4$ NPs decreased to 1.3%, which indicated that there were many carboxyl groups grafted on the surface of Fe$_3$O$_4$ NPs. The iron content on the surface of the other three

modified NPs was similar, indicating that the number of grafted groups had no significant difference on the surface of $Fe_3O_4$@OH, $Fe_3O_4$@COOH(OA) and $Fe_3O_4$@$NH_2$ NPs.

The above results indicate that the four modification processes did not have a significant influence on the particle size of $Fe_3O_4$ NPs. Among them, the modification processes of dextran and APTES had no significant effect on the super-paramagnetism of $Fe_3O_4$ NPs.

### 3.1.5. Zeta potential analysis

Zeta potential is the electrostatic potential that exists on the shear plane of the particle and is related to the surface charge and the local environment of the particle. As shown in figure 4, the charge of the particles before and after modification was significantly different; $Fe_3O_4$ and modified $Fe_3O_4$ have a positive charge when the pH is less than 3.5. However, when the pH is between 4 and 10, $Fe_3O_4$@$NH_2$ carries a negative charge, while $Fe_3O_4$ and other modified $Fe_3O_4$ MNPs carry a positive charge. APTES changed the electrification state of $Fe_3O_4$ at pH = 4–10, and it was also found that the electric potential value of $Fe_3O_4$@$NH_2$ MNPs was significantly greater than $Fe_3O_4$. The isoelectric value of $Fe_3O_4$@$NH_2$ MNPs was significantly changed to 9–9.5. Therefore, $Fe_3O_4$@$NH_2$ MNPs carry a positive charge in a wide pH range of 3.0–10.0, which is due to the ionization of $NH_2$, the functional group of APTES.

The cell wall of Gram-positive bacteria (G+) contains a thick layer (15–30 nm) of peptidoglycan, attached to the phosphopartic acid, and has no outer membrane, so there is no obvious periplasmic space. Peptidoglycan layer and phosphoparic acid together form a polyanionic matrix with various functions. The surface components attached to the cell wall contribute to the net negative charge of the cell. Over a wide range of pH, bacteria carry a negative charge on the surface of their cells, which can attract positively charged ones by electrostatic attraction. The APTES functional group enhanced the electrostatic adsorption of bacteria to the $Fe_3O_4$ MNPs.

## 3.2. Interaction between $Fe_3O_4$ MNPs and bacteria cells

### 3.2.1. Conditions for immobilization of cells and particles

To investigate the optimum incubation time for the immobilization of cells, a growth curve analysis of cell density and cell weight was performed, as shown in figure 5a. During the 28 h incubation period, the cell growth reached the post-log phase with $OD_{600}$ of 0.593 and DCW of 18.3 g $l^{-1}$. According to a previous study, after incubation for 28 h, the cells collected from the growth medium were resuspended to an $OD_{600}$ of 0.2, and the weight of the NPs for immobilization was twice the DCW. Briefly, in the current study, after cultivation of the cells for 28 h, cells from the 10 ml LB fermented liquid were diluted to 30 ml with the mineral medium, and 0.366 g of $Fe_3O_4$ NPs were added for immobilization at 4°C and vibration for 12 h.

### 3.2.2. Effects of ethanol treatment on bacterial capture of $Fe_3O_4$ magnetic nanoparticles

*Rhodococcus erythropolis* SY095, a Gram-positive bacterium, has no lipid membrane in its cell wall. The outermost layer is a thick peptidoglycan layer composed of N-acetylglucosamine and N-acetylcytic acid, on which phosphopartic acid is distributed. *Rhodococcus erythropolis* SY095 has a unique lipophilic surface structure and abundant fatty acid chain on the surface. Ethanol lyses fatty acid chains on the surface of cells, it can be seen from figure 5b that the adsorption of $Fe_3O_4$ MNPs on cells treated with ethanol was significantly reduced, indicating that the part of the adsorption between $Fe_3O_4$ MNPs and bacteria was through the interaction between fatty acid chains on the surface of bacteria and particles.

### 3.2.3. Analysis of links between bacteria and particles

This study speculated that MNPs with bacterial surface mainly through the following three ways: (i) It can be seen from §3.1.5, Zeta potential analysis, the functional group $NH_2$ modified on the surface of $Fe_3O_4$ MNPs significantly changed the isoelectric point of MNPs, which made it easier for bacteria to capture particles on their surfaces. Many positive changes introduced by APTES for $Fe_3O_4$ play an important role in the interaction, so $Fe_3O_4$@$NH_2$ MNPs can interact directly with bacterial cells by electrostatic forces. (ii) Hydroxyl or carboxylated MNPs ionize H+, which made their surface negatively charged and easy to bind to amino proteins on the cell surface, the interaction between them was mainly through the formation of electrostatic adsorption between carboxyl or hydroxyl and amino groups. (iii) *Rhodococcus erythropolis* has a special surface structure and the fatty acid chain on the surface can easily capture MNPs, which is shown in figure 6a,b. It was confirmed in §3.2.2 that

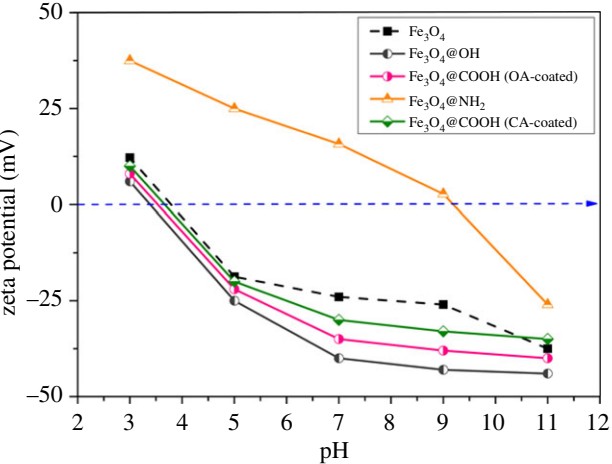

**Figure 4.** Zeta potential is compared between $Fe_3O_4$ and modified magnetic $Fe_3O_4$ NPs when pH = 3–11.

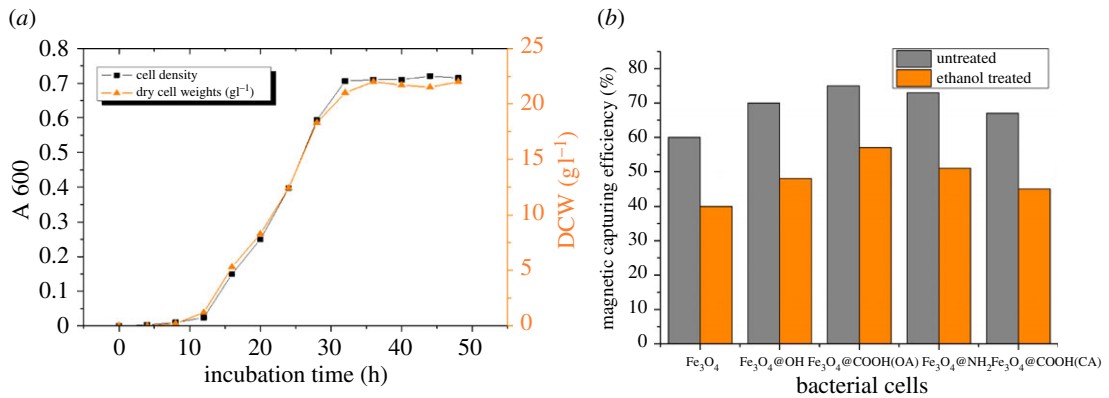

**Figure 5.** (*a*) Growth curve of *R. erythropolis* SY095: cell density and DCW. (*b*) Magnetic capturing efficiency of cells treated with and without ethanol by $Fe_3O_4$ and modified magnetic $Fe_3O_4$ NPs.

the capture rate of MNPs decreased significantly after ethanol treatment, which suggested that the binding of bacteria to MNPs was related to the coordination of fatty acid chain on the surface of bacteria.

## 3.3. Degradation of hexadecane and crude oil by the immobilized cells: SY095-$Fe_3O_4$ MNPs

We compared the degradation of hexadecane by free bacterial cells and immobilized bacterial cells (figure 7*a*). The immobilized cells showed activity similar to that of the free cells. Moreover, degradation by $Fe_3O_4$@COOH/OA NP-coated cells was higher than that of the free cells. Both could completely degrade 100 mg l$^{-1}$ hexadecane after 72 h of culture, which shows that there was no mass transfer obstacle in the system.

Properties of the immobilized cells with different functional $Fe_3O_4$ NPs on secondary oil recovery wastewater degradation were also compared. Figure 7*b* shows that in the initial stages, three immobilized cell types (i.e. coated with $Fe_3O_4$@COOH(CA), $Fe_3O_4$@OH and $Fe_3O_4$@NH$_2$) exhibited degradation, but their performance was slightly lower compared with that of the free cells; the degradation rate of the immobilized cells was similar to that of the free cells in the later period of the culture. Immobilized cells coated with OA had a degradation rate higher than that of the free cells in the initial stage of the culture. This may be due to the lipophilic nature of the alkyl chain of OA, which could have made it easier for the cells to absorb oils. In the middle stages of the culture, cells could secrete surface-active substances, and the free cells had a degradation rate similar to that of the immobilized cells (coated with $Fe_3O_4$@COOH(OA)). In conclusion, the immobilized cells had a slightly lower degradation rate than that of the free cells. Degradation rates of the immobilized cells coated with $Fe_3O_4$@OH were the closest to that of the free cells. The surface-modified $Fe_3O_4$ NPs did not affect the degradation activity of the cells.

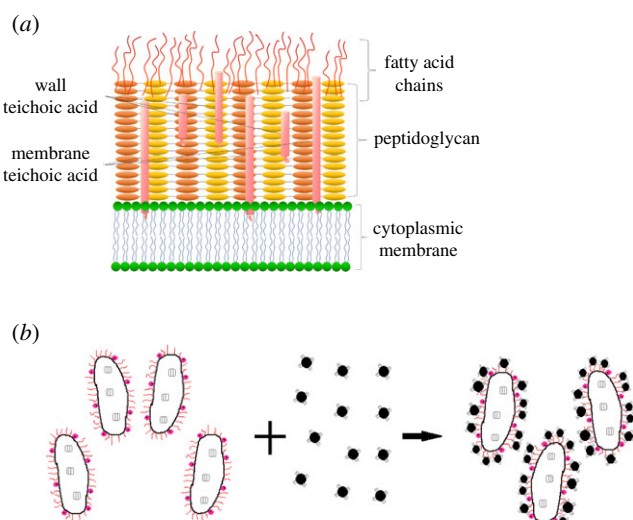

**Figure 6.** (*a*) Surface structure of *R. erythropolis* cell. (*b*) Link between *R. erythropolis* cell and MNP.

## 3.4. Effect of different heavy metals on the degradation of secondary oil resuming wastewater

We also studied the tolerance of the bacteria toward heavy metals for practical application. Figure 8 shows that $Cr^{2+}$ and $Pb^{2+}$ had different inhibitory impacts on the activity of the free cells, and a crude oil degradation efficiency of 27% and 45%, respectively. Interestingly, no inhibition was observed after the free cells were coated with MNPs. However, it was observed that the reducibility of all $Cr^{2+}$-containing crude oil samples was higher as compared with that of the control group. MNPs probably promote the activity of the related degradation enzymes. In addition, the crude oil reduction capability of the coated MNPs increased (@COOH/CA) 2-, (@COOH/OA) 2.1-, (@OH) 2.2- and (@NH$_2$) 1.7-folds after the addition of $Cr^{2+}$. Thus, these results indicate that these heavy metals greatly affected the degradation activity of *R. erythropolis* SY095, which was protected by MNPs after immobilization. Among them, $Fe_3O_4$@OH NPs had the strongest protective effect on the bacterial cells.

## 3.5. Analysis of hexadecane and crude oil degradation products in immobilized cells SY095-modified $Fe_3O_4$ NPs

We analysed hexadecane degradation products in the immobilized cell culture via GC/MS. Table 2 and electronic supplementary material, S.1 show that one of the compounds was identified as hexadecane, which was also detected in the case of free cells. Table 3 shows that the free cells had slightly smaller peak areas than that of some of the immobilized cells with no significant change. Furthermore, immobilized cells coated with $Fe_3O_4$@OH NPs showed better degradation than the free cells. Hence, the results showed that the degradation pathway of cetane was the same as that of the immobilized SY095 cells with functional $Fe_3O_4$ NPs. $Fe_3O_4$@OH NPs coated on the bacterial surface could enhance the metabolism of alkanes.

GC/MS was also used for analysing the crude oil degradation intermediates during degradation analysis with the immobilized cells. Table 3 shows that six main compounds of crude oil were identified in the analysis. The same metabolites were detected in the case of both free cells and immobilized cells. Hence, these results suggest that crude oil did not change its degradation pathway after the immobilization of the SY095 strain with functional $Fe_3O_4$ NPs. The predicted chemical structure, retention time [11], as well as peak area is summarized in table 3. Furthermore, crude oil degradation by the immobilized cells had the same metabolites as those by the free cells. As indicated by the peak area (table 3), the degradation effects of the immobilized cells were similar to that of the free cells. The coated $Fe_3O_4$ NPs did not affect the original degradation pathway and mass transfer of the cell membrane. Moreover, the degradation efficiency of the bacterial cells immobilized by $Fe_3O_4$@OH was better than that of the free cells, which may be because of the protective effect of the coated groups on the cell surface.

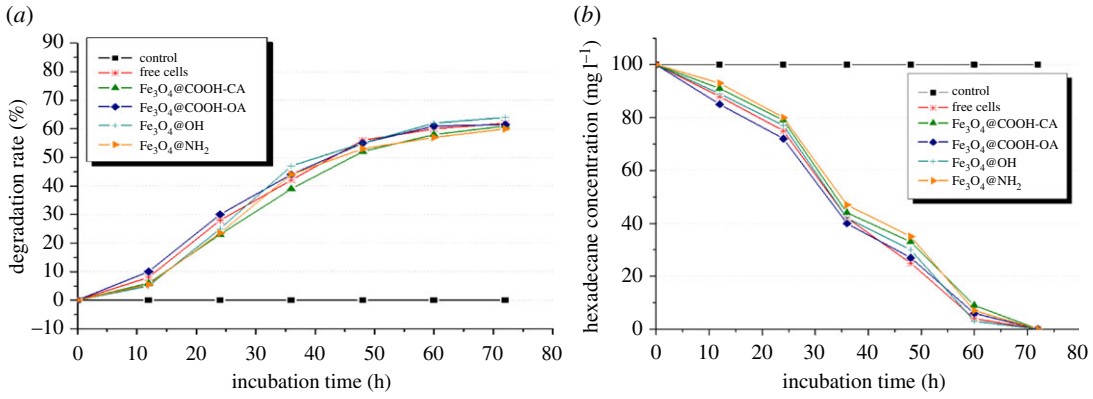

**Figure 7.** Degradation of hexadecane and crude oil by the free bacterial cells and immobilized bacterial cells. (*a*) Time course of free and $Fe_3O_4$ NP-coated cells in decreasing oil. Immobilized cells and $Fe_3O_4$@COOH/OA NP-coated cells could completely degrade 100 mg l$^{-1}$ hexadecane after 72 h of culture. (*b*) Comparing immobilized cells with different functional $Fe_3O_4$ NPs on secondary oil recovery wastewater degradation. Time course of cetane reduction in free and $Fe_3O_4$ NP-coated cells. In the initial stages, three immobilized cell types (i.e. coated with $Fe_3O_4$@COOH(CA), $Fe_3O_4$@OH, and $Fe_3O_4$@NH$_2$) exhibited degradation. Control: $Fe_3O_4$ NPs without cells; free cells: cells without $Fe_3O_4$ NPs. Initial hexadecane concentration: 225 mg l$^{-1}$; incubation: 72 h.

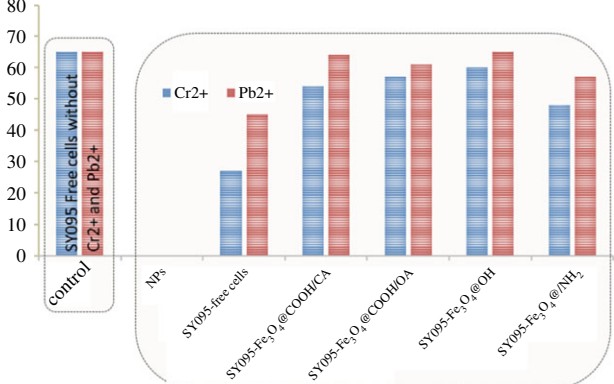

**Figure 8.** Degradation of SY095 in the presence of heavy metals. Control: SY095 free cells without heavy metals; NPs: $Fe_3O_4$ NPs without cells; Cr2+: SY095 free cells or immobilized cells with Cr2+; Pb2+: SY095 free cells or immobilized cells with Pb2+. Initial total crude oil concentration: 225 mg l$^{-1}$; incubation: 72 h.

**Table 2.** Properties of degradation products of cetane determined by GC/MS.

| retention time | chemical structure | peak area | | | | | |
| --- | --- | --- | --- | --- | --- | --- | --- |
| | | control[a] | free cells | $Fe_3O_4$@COOH-CA | $Fe_3O_4$@COOH-OA | $Fe_3O_4$@OH | $Fe_3O_4$@ NH$_2$ |
| 28.293 | $C_6H_{14}$ | $4.448 \times 10^{10}$ | $2.567 \times 10^{10}$ | $2.592 \times 10^{10}$ | $2.621 \times 10^{10}$ | $2.375 \times 10^{10}$ | $2.684 \times 10^{10}$ |

[a]Control: $Fe_3O_4$ NPs without cells. The initial concentration of hexadecane was 1000 mg l$^{-1}$. HP-5 MS column using helium as carrier gas and 1.0 ml min$^{-1}$ flow rate (US, 30 m $\times$ 250 mm $\times$ 0.25 mm); injector temperature: 230℃; interface temperature and ion source temperature: 250℃; column temperature was increased from 100℃ for 1 min, raised at 2℃ min$^{-1}$ to 200℃, after that increased at 10℃ min$^{-1}$ to 300℃ (for 5 min). Incubation for 36 h.

## 3.6. Repeated use of magnetically immobilized cells SY095-$Fe_3O_4$@OH NPs for secondary oil recovery biodegradation

Under the same conditions, we were able to isolate the magnetically immobilized cells from the culture medium and use them continuously in the reaction mixture for at least five cycles (figure 9). The results showed that after five cycles, the enveloped cells still had higher activity than the original enveloped cells (greater than or equal to 90%). However, as the reaction period increased, the degradation rate of crude

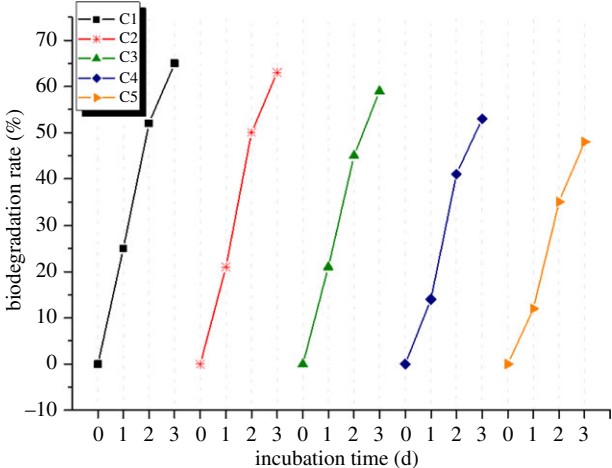

**Figure 9.** Reusability and stability of coated-SY095 cells for degradation of crude oil SY095 cells coated with $Fe_3O_4$@OH NPs; incubation: 72 h for every cycle. Electronic supplementary material, S.1 shows gas chromatography analysis of cetane degradation by immobilized cells in culture medium. Control: $Fe_3O_4$ NPs without cells. Initial hexadecane concentration: 1000 mg $l^{-1}$; incubation: 36 h.

**Table 3.** Properties of crude oil degradation products determined by GC/MS.

| retention time | chemical mix | peak area | | | | | |
|---|---|---|---|---|---|---|---|
| | | control[a] | free cells | $Fe_3O_4$@COOH-CA | $Fe_3O_4$@COOH-OA | $Fe_3O_4$@OH | $Fe_3O_4$@ $NH_2$ |
| 21.477 | $C_{10}H_{22}O_3S_2$ | $2.499 \times 10^8$ | — | — | — | — | — |
| 22.669 | $C_{12}H_{25}F$ | $6.369 \times 10^8$ | $7.711 \times 10^7$ | $1.146 \times 10^8$ | $1.245 \times 10^8$ | $6.168 \times 10^7$ | $1.680 \times 10^8$ |
| 24.815 | $C_{21}H_{44}$ | $6.301 \times 10^8$ | $1.012 \times 10^8$ | $1.389 \times 10^8$ | $1.669 \times 10^8$ | $8.464 \times 10^7$ | $2.321 \times 10^8$ |
| 35.482 | $C_{23}H_{32}O_2$ | $8.527 \times 10^8$ | $5.319 \times 10^8$ | $5.207 \times 10^8$ | $5.186 \times 10^8$ | $5.074 \times 10^8$ | $8.425 \times 10^8$ |
| 41.339 | $C_{15}H_{26}O$ | $2.117 \times 10^8$ | $1.031 \times 10^8$ | $1.148 \times 10^8$ | $1.001 \times 10^8$ | $9.463 \times 10^7$ | $1.073 \times 10^8$ |
| 41.950 | $C_{17}H_{30}O_3$ | $3.044 \times 10^8$ | $1.134 \times 10^8$ | $1.132 \times 10^8$ | $1.552 \times 10^8$ | $1.124 \times 10^8$ | $1.275 \times 10^8$ |

[a]Control: $Fe_3O_4$ NPs without cells. The initial concentration of total crude oil was 1000 mg $l^{-1}$. HP-5 MS column using helium as carrier gas and 1.0 ml $min^{-1}$ flow rate (US, 30 m $\times$ 250 mm $\times$ 0.25 mm); injector temperature: 230℃; interface temperature and ion source temperature: 250℃; column temperature was increased from 100℃ for 1 min, raised at 2℃ $min^{-1}$ to 200℃, after that increased at 10℃ $min^{-1}$ to 300℃ (for 5 min). Incubation for 72 h.

oil decreased significantly. In the fifth reaction cycle, the 3-day degradation rate was only approximately 50%; this may be because the coated cells were gradually being lost after magnetic separation of each cycle. These results showed that the $Fe_3O_4$@OH NP-immobilized cells could maintain degradation activity for less than three cycles.

Previously, we screened and maintained a hexadecane degrading strain (CGMCC no. 10724)—*R. erythropolis* SY095, which could similarly use crude oil as a carbon source. A previous study investigated the influence of the surfactant on the solubility of n-hexadecane, the efficiency of n-hexadecane degradation, the proliferation of bacteria and the bacteria's hydrophobicity [49]. Additionally, another study investigated the removal of lead and copper from soils using SY095 biosurfactants [50]. Earlier research has concentrated on *Erythrococcus*' extracellular products rather than its breakdown capabilities. However, the current strain will be used as a substrate for MNP coating, and the effect of MNPs with different surface groups on the biodegradation ability of SY095 as well as whether they have a protective effect against heavy metal toxicity will be investigated.

## 4. Conclusion

Crude oil is a complex mixture containing many thousands of different hydrocarbon compounds. Alkali-surfactant-polymer flooding technology is widely employed to extract crude oil to enhance

its production. The bacterial strain *R. erythropolis* SY095 has shown high degradation activity of alkane of crude oil. In this study, a new immobilization technology is proposed based on nano functional $Fe_3O_4$ NPs, advantages of magnetic nanomaterials and microbial biodegradation. *Rhodococcus erythropolis* SY095 was successfully coated with functional MNPs that could be easily separated from the solution via the application of an external magnetic field. The coated cells had a high degradation rate and a high tolerance for heavy metals. We found that the bacteria coated with functional $Fe_3O_4$ NPs displayed higher degradation rates for hexadecane and crude oil than that displayed by the free bacterial cells. Given the insolubility of alkanes in water, and their low biological availability to bacteria, the biodegradation in bacterial cultures with alkanes as a sole source of carbon triggers the stationary phase. Due to the immobilization of the bacterial cells with MNPs, the degradation rates increased. The immobilized cells had a higher degradation rate than that of the free cells even in the presence of heavy metals ($Cr^{2+}$, $Pb^{2+}$). *Rhodococcus erythropolis* SY095 showed strong biodegradation ability and enhanced tolerance to heavy metal toxicity after being wrapped by MNPs during crude oil degradation. The participation of NPs had no change in the transformation pathway. Our findings indicate that immobilizing MNPs on bacterial surfaces is a potential strategy for crude oil degradation.

## 4.1. Limitation and future perspective

There are several limitations to our study. We do not study the properties of immobilizing MNPs based on the temperature, pH, salinity and other environmental conditions. Therefore, more studies are needed to study the current method on diverse temperature, pH, salinity and other environmental conditions. Moreover, the current study lacks vigorous toxicity studies, which need to be evaluated in future studies. Future studies should use a sustainable and environmentally friendly method as an alternative to traditional methods. Moreover, cost-effective approaches should be considered in future studies which can exhibit practical applications.

Data accessibility. The datasets that support the results of this article are present in the article and its attached files.
Authors' contributions. X.M. contributed to the conception of the study and performed the experiment and wrote the manuscript; B.X. and D.D. contributed significantly to analysis of data; X.W. contributed manuscript preparation; J.C. helped perform the analysis with constructive discussions. All authors read and approved the manuscript. All authors gave final approval for publication and agreed to be held accountable for the work performed therein.
Competing interests. The authors have no conflict of interests to declare.
Funding. This work was supported by National Key Research and Development Program of China (grant no. 2018YFE0196000), Tianjin Science and Technology Planning Project (grant no. 20JCYBJC00830) and Basic Research Fund of The Central Public Welfare Scientific Research Institutions (grant no. K-JBYWF-2019-T06).

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
