## [Peer Review File · Royal Society Open Science]

Review History

RSOS-210135.R0 (Original submission)

Review form: Reviewer 1

Is the manuscript scientifically sound in its present form?

No

Are the interpretations and conclusions justified by the results?

No

Is the language acceptable?

No

Do you have any ethical concerns with this paper?

Yes

Have you any concerns about statistical analyses in this paper?

No

Recommendation?

Reject

Comments to the Author(s)

Dear Author

Hi!

I checked manuscript entitled "Degradation of Rhodococcus erythropolis SY095 modified with functional magnetic Fe₃O₄ nanoparticles". This manuscript is not suitable for publication in Royal Society. In my opinion above title is suitable for specialized journal.

Regards

Sincerely,

Review form: Reviewer 2**Is the manuscript scientifically sound in its present form?**

Yes

Are the interpretations and conclusions justified by the results?

Yes

Is the language acceptable?

No

Do you have any ethical concerns with this paper?

No

Have you any concerns about statistical analyses in this paper?

No

Recommendation?

Major revision is needed (please make suggestions in comments)

Comments to the Author(s)

The authors reported Surface modification and functionalization of Rhodococcus erythropolis SY095 using magnetic Fe₃O₄ NPs for improving biodegradation of hexadecane and crude oil, the investigation presented in the manuscript are quite new, however I recommend the major revision based on the following comments.

The authors stated in Line 73: "Previously, we screened a hexadecane degrading strain – Rhodococcus erythropolis 74 SY095, which could also use crude oil as a carbon source". The authors should complete the sentence with reference and explain the results briefly to describe the difference with the present work.

The authors used Fe₃O₄ NPs for modifying the surface of Rhodococcus erythropolis SY095, the authors should explain more detail the purpose of the use of some kinds of MNPs such as hydrophilic Fe₃O₄@COOH, lipophilic Fe₃O₄@COOH and Fe₃O₄@OH.

Line 136: "The cells were coated with modified Fe₃O₄ NPs" Experimental section is not written well enough It should be in a way that the coating process can be reproducible even for non-expert researcher.

In the XRD patterns, the authors should identify each the diffraction peak with miller index or phases.

In the hysteresis loops, it is difficult to distinguish the loop for each sample, the authors should modify Fig 2(b).

The explanation of Ms modification of MNPs are insufficient. The authors should explain more detail, why Ms change.

Decision letter (RSOS-210135.R0)

Dear Dr Ma:

Manuscript ID: RSOS-210135

Title: "Degradation of Rhodococcus erythropolis SY095 modified with functional magnetic Fe₃O₄ nanoparticles"

Thank you for submitting the above manuscript to Royal Society Open Science. Your paper was sent to reviewers and their comments are included at the bottom of this letter. I apologise for the delay in sending you this decision.

In view of the concerns raised by the reviewers, the manuscript has been rejected in its current form. However, a new manuscript may be submitted which takes into consideration these comments.

Please note that resubmitting your manuscript does not guarantee eventual acceptance, and that your resubmission will be subject to peer review before a decision is made.

Your resubmitted manuscript should be submitted by 08-Dec-2021. If you are unable to submit by this date please contact the Editorial Office.

Royal Society of Chemistry
Thomas Graham House

Science Park, Milton Road
Cambridge, CB4 0WF
Royal Society Open Science - Chemistry Editorial Office

On behalf of the Subject Editor Professor Anthony Stace and the Associate Editor Dr Nadia Martinez Villegas

REVIEWER(S) REPORTS:

Associate Editor Comments to Author ():

RSC Associate Editor:

Comments to the Author:

The research presented in this draft is original and might be of interest to RSOS audience. However redrafting of the manuscript is needed in order to suit RSOS audience. Additionally, more details are needed to improve the Material and Methods section and achieve a scientifically sound experimental design. For doing so, please see comments from the reviewer.

RSC Subject Editor:

Comments to the Author:

(There are no comments.)

Reviewers' Comments to Author:

Reviewer: 1

Comments to the Author(s)

Dear Author

Hi!

I checked manuscript entitled "Degradation of Rhodococcus erythropolis SY095 modified with functional magnetic Fe₃O₄ nanoparticles". This manuscript is not suitable for publication in Royal Society. In my opinion above title is suitable for specialized journal.

Regards

Sincerely,

Reviewer: 2

Comments to the Author(s)

The authors reported Surface modification and functionalization of Rhodococcus erythropolis SY095 using magnetic Fe₃O₄ NPs for improving biodegradation of hexadecane and crude oil, the investigation presented in the manuscript are quite new, however I recommend the major revision based on the following comments.

The authors stated in Line 73: "Previously, we screened a hexadecane degrading strain – Rhodococcus erythropolis 74 SY095, which could also use crude oil as a carbon source". The authors should complete the sentence with reference and explain the results briefly to describe the difference with the present work.

The authors used Fe₃O₄ NPs for modifying the surface of Rhodococcus erythropolis SY095, the authors should explain more detail the purpose of the use of some kinds of MNPs such as hydrophilic Fe₃O₄@COOH, lipophilic Fe₃O₄@COOH and Fe₃O₄@OH.

Line 136: "The cells were coated with modified Fe₃O₄ NPs" Experimental section is not written well enough It should be in a way that the coating process can be reproducible even for non-expert researcher.

In the XRD patterns, the authors should identify each the diffraction peak with miller index or phases.

In the hysteresis loops, it is difficult to distinguish the loop for each sample, the authors should modify Fig 2(b).

The explanation of Ms modification of MNPs are insufficient. The authors should explain more detail, why Ms change.

Author's Response to Decision Letter for (RSOS-210135.R0)

See Appendix A.

RSOS-211172.R0

Review form: Reviewer 1

Is the manuscript scientifically sound in its present form?

No

Are the interpretations and conclusions justified by the results?

No

Is the language acceptable?

No

Do you have any ethical concerns with this paper?

No

Have you any concerns about statistical analyses in this paper?

No

Recommendation?

Major revision is needed (please make suggestions in comments)

Comments to the Author(s)

Title: Degradation of Rhodococcus erythropolis SY095 modified with functional magnetic Fe₃O₄ nanoparticles

Dear Editor/Editor-in-Chief

The topic is interesting and within the scope of the journal. After major revision, it may be suitable for publication.

- 1 Abstract has not started with appropriate sentences and it is recommended that the basic sentences of Abstract be improved.
- 2 There are a lot of grammatical, typographical, and punctuation errors in this manuscript that should be carefully checked and corrected.
- 3 There is no enough background given to understand the rationale. The Introduction part is not very well-structured and lacks a clear backbone.
- 4 The figure 3 (SEM) is not clear and need to be replaced by figures with higher resolution
- 5 For table 2 and 3 , the conditions should be explain completely. Explain more and in detail.
- 6 The Fig 1 (FTIR) is important and has not been described completely. Explain it more.
- 7 Table 1, the numbers should be rounded to two decimal places.
- 8 Complete and acceptable explanations and interpretations should be provided for fig 7.
- 9 The conclusion section should be improved. Conclusions should be supported by the data.

In the introductory section, the author is suggested to refer the cited articles. At the end of the text: " However, the synthesis of MNPs with consistent shapes and proven dispersion remains a challenge."

It is suggested to refer to the following article:

[1] R. Jalilian and A. Taheri, Synthesis and application of a novel core-shell-shell magnetic ion imprinted polymer as a selective adsorbent of trace amounts of silver ions. *e-Polymers*, 18 (2018) 123-134. DOI: 10.1515/epoly-2017-0108

[2] A. Talavari, B. Ghanavati, A. Azimi and S. Sayyahi, PVDF/ MWCNT hollow fiber mixed matrix membranes for gas absorption by Al₂O₃ nanofluid *Progress in Chemical and Biochemical Research*, 4 (2021) 177-190. DOI: 10.22034/pcbr.2021.270178.1177

[3] M. Shahamatpour, S.M. Tabatabaee Ghomsheh, S. Maghsoudi and S. Azizi, Fenton Processes, Adsorption and Nano Filtration in Industrial Wastewater Treatment. *Progress in Chemical and Biochemical Research*, 4 (2021) 31-43. DOI: 10.22034/pcbr.2021.118152

[4] Z. Rezayati zad, B. Moosavi and A. Taheri, Synthesis of monodisperse magnetic hydroxyapatite/Fe₃O₄ nanospheres for removal of Brilliant Green (BG) and Coomassie Brilliant Blue (CBB) in the single and binary systems. *Advanced Journal of Chemistry-Section B*, 2 (2020) 159-171. DOI: 10.33945/sami/ajcb.2020.3.8

[5] H. Thacker, V. Ram and P.N. Dave, Plant mediated synthesis of Iron nanoparticles and their Applications: A Review. *Progress in Chemical and Biochemical Research*, 2 (2019) 84-91. DOI: DOI: 10.33945/SAMI/PCBR.183239.1033

- 10 Please compare the performance of this method with some previously reported works.
- 11 From my perspective of view, the present work is far from satisfaction. I would suggest the authors to include the drawbacks and possible future work for improvement in the conclusion part.
- 12 Add some references related to the 2021 or 2020 research topic.

Review form: Reviewer 2

Is the manuscript scientifically sound in its present form?

Yes

Are the interpretations and conclusions justified by the results?

Yes

Is the language acceptable?

Yes

Do you have any ethical concerns with this paper?

No

Have you any concerns about statistical analyses in this paper?

No

Recommendation?

Accept with minor revision (please list in comments)

Comments to the Author(s)

Ms of Fe₃O₄ was 38.2. Basically, Ms of Fe₃O₄ coated by non-magnetic material will decrease. However, the authors reported Ms of Fe₃O₄@COOH (OA-coated) slightly increased. Why?

Decision letter (RSOS-211172.R0)

Dear Dr Ma:

Title: Degradation of Rhodococcus erythropolis SY095 modified with functional magnetic Fe₃O₄ nanoparticles

Manuscript ID: RSOS-211172

The editor assigned to your paper has now received comments from reviewers. We would like you to revise your paper in accordance with the referee and Subject Editor suggestions which can be found below (not including confidential reports to the Editor). Please note this decision does not guarantee eventual acceptance.

Please submit a copy of your revised paper before 07-Oct-2021. Please note that the revision deadline will expire at 00.00am on this date. If we do not hear from you within this time then it will be assumed that the paper has been withdrawn. In exceptional circumstances, extensions may be possible if agreed with the Editorial Office in advance. We do not allow multiple rounds of revision so we urge you to make every effort to fully address all of the comments at this stage. If deemed necessary by the Editors, your manuscript will be sent back to one or more of the original reviewers for assessment. If the original reviewers are not available we may invite new reviewers.

To revise your manuscript, log into <http://mc.manuscriptcentral.com/rsos> and enter your Author Centre, where you will find your manuscript title listed under "Manuscripts with

Decisions." Under "Actions," click on "Create a Revision." Your manuscript number has been appended to denote a revision. Revise your manuscript and upload a new version through your Author Centre.

Yours sincerely,
Dr Ellis Wilde
Publishing Editor, Journals

On behalf of the Subject Editor Professor Anthony Stace and the Associate Editor Dr Nadia Martinez Villegas.

RSC Associate Editor

Comments to the Author:

The research presented in this draft is original and might be of interest to RSOS audience. However, the abstract and the introduction need to be substantially improved. Additionally, any statistics to help support a high quality assurance and control must be included. Error bars and statistical differences should be indeed added to all relevant figures. Finally, the English language and punctuation of this manuscript should be revised.

Reviewers' Comments to Author:

Reviewer: 1

Comments to the Author(s)

Title: Degradation of Rhodococcus erythropolis SY095 modified with functional magnetic Fe₃O₄ nanoparticles

Dear Editor/Editor-in-Chief

The topic is interesting and within the scope of the journal. After major revision, it may be suitable for publication.

1 Abstract has not started with appropriate sentences and it is recommended that the basic sentences of Abstract be improved.

- 2 There are a lot of grammatical, typographical, and punctuation errors in this manuscript that should be carefully checked and corrected.
- 3 There is no enough background given to understand the rationale. The Introduction part is not very well-structured and lacks a clear backbone.
- 4 The figure 3 (SEM) is not clear and need to be replaced by figures with higher resolution
- 5 For table 2 and 3 , the conditions should be explain completely. Explain more and in detail.
- 6 The Fig 1 (FTIR) is important and has not been described completely. Explain it more.
- 7 Table 1, the numbers should be rounded to two decimal places.
- 8 Complete and acceptable explanations and interpretations should be provided for fig 7.
- 9 The conclusion section should be improved. Conclusions should be supported by the data.

In the introductory section, the author is suggested to refer the cited articles. At the end of the text:“ However, the synthesis of MNPs with consistent shapes and proven dispersion remains a challenge.”

It is suggested to refer to the following article:

[1] R. Jalilian and A. Taheri, Synthesis and application of a novel core-shell-shell magnetic ion imprinted polymer as a selective adsorbent of trace amounts of silver ions. *e-Polymers*, 18 (2018) 123-134. DOI: 10.1515/epoly-2017-0108

[2] A. Talavari, B. Ghanavati, A. Azimi and S. Sayyahi, PVDF/ MWCNT hollow fiber mixed matrix membranes for gas absorption by Al₂O₃ nanofluid *Progress in Chemical and Biochemical Research*, 4 (2021) 177-190. DOI: 10.22034/pcbr.2021.270178.1177

[3] M. Shahamatpour, S.M. Tabatabaee Ghomsheh, S. Maghsoudi and S. Azizi, Fenton Processes, Adsorption and Nano Filtration in Industrial Wastewater Treatment. *Progress in Chemical and Biochemical Research*, 4 (2021) 31-43. DOI: 10.22034/pcbr.2021.118152

[4] Z. Rezayati zad, B. Moosavi and A. Taheri, Synthesis of monodisperse magnetic hydroxyapatite/Fe₃O₄ nanospheres for removal of Brilliant Green (BG) and Coomassie Brilliant Blue (CBB) in the single and binary systems. *Advanced Journal of Chemistry-Section B*, 2 (2020) 159-171. DOI: 10.33945/sami/ajcb.2020.3.8

[5] H. Thacker, V. Ram and P.N. Dave, Plant mediated synthesis of Iron nanoparticles and their Applications: A Review. *Progress in Chemical and Biochemical Research*, 2 (2019) 84-91. DOI: DOI: 10.33945/SAMI/PCBR.183239.1033

- 10 Please compare the performance of this method with some previously reported works.
- 11 From my perspective of view, the present work is far from satisfaction. I would suggest the authors to include the drawbacks and possible future work for improvement in the conclusion part.
- 12 Add some references related to the 2021 or 2020 research topic.

Reviewer: 2

Comments to the Author(s)

Ms of Fe₃O₄ was 38.2. Basically, Ms of Fe₃O₄ coated by non-magnetic material will decrease. However, the authors reported Ms of Fe₃O₄@COOH (OA-coated) slightly increased. Why?

Author's Response to Decision Letter for (RSOS-211172.R0)

See Appendix B.

RSOS-211172.R1

Review form: Reviewer 1

Is the manuscript scientifically sound in its present form?

Yes

Are the interpretations and conclusions justified by the results?

Yes

Is the language acceptable?

Yes

Do you have any ethical concerns with this paper?

No

Have you any concerns about statistical analyses in this paper?

No

Recommendation?

Accept as is

Comments to the Author(s)

The authors have addressed satisfactorily all my comments on the R1 version of this manuscript. Therefore, I recommend that the paper is accepted for publication.

Decision letter (RSOS-211172.R1)

Dear Dr Ma:

Title: Degradation of *Rhodococcus erythropolis* SY095 modified with functional magnetic Fe₃O₄ nanoparticles

Manuscript ID: RSOS-211172.R1

It is a pleasure to accept your manuscript in its current form for publication in Royal Society Open Science. The chemistry content of Royal Society Open Science is published in collaboration with the Royal Society of Chemistry.

Please see the Royal Society Publishing guidance on how you may share your accepted author manuscript at <https://royalsociety.org/journals/ethics-policies/media-embargo/>. After publication, some additional ways to effectively promote your article can also be found here

<https://royalsociety.org/blog/2020/07/promoting-your-latest-paper-and-tracking-your-results/>.

Yours sincerely,
Dr Ellis Wilde
Publishing Editor, Journals

On behalf of the Subject Editor Professor Anthony Stace and the Associate Editor Dr Nadia Martinez Villegas.

RSC Associate Editor
Comments to the Author:
The authors have addressed all referee comments up to their satisfaction. Therefore, the paper can be accepted for publication.

RSC Subject Editor
Comments to the Author:
(There are no comments.)

Reviewer(s)' Comments to Author:
Reviewer: 1
Comments to the Author(s)
The authors have addressed satisfactorily all my comments on the R1 version of this manuscript. Therefore, I recommend that the paper is accepted for publication.

Appendix A

Comments to the Author(s)

The authors reported Surface modification and functionalization of *Rhodococcus erythropolis* SY095 using magnetic Fe₃O₄ NPs for improving biodegradation of hexadecane and crude oil, the investigation presented in the manuscript are quite new, however I recommend the major revision based on the following comments.

- ✓ The authors stated in Line 73: “Previously, we screened a hexadecane degrading strain—*Rhodococcus erythropolis* 74 SY095, which could also use crude oil as a carbon source”. The authors should complete the sentence with reference and explain the results briefly to describe the difference with the present work.

Line 90-100.

Previously, we screened a hexadecane degrading strain and applied culture preservation in China General Microbiological Culture Collection Center for the strain (CGMCC No.10724)—*Rhodococcus erythropolis* SY095, which could also use crude oil as a carbon source. In our previous studies, the effects of the surfactant on the solubility of n-hexadecane, the degradation efficiency of n-hexadecane, the growth of the bacteria and the hydrophobicity of the bacteria were studied [39]. In addition, we also studied the removal of lead and copper from sediments by biosurfactants from SY095 [40]. Previous studies mainly focused on the extracellular products of *erythroccoccus* rather than its degradation properties. Therefore, in this study, this strain will be used as substrate for magnetic nanoparticle coating, and the influence of magnetic nanoparticle with different surface groups on the biodegradation ability of SY095 will be studied, as well as whether it has protective effect on SY095 from heavy metal toxicity.

- ✓ The authors used Fe₃O₄ NPs for modifying the surface of *Rhodococcus erythropolis* SY095, the authors should explain more detail the purpose of the use of some kinds of MNPs such as hydrophilic Fe₃O₄@COOH, lipophilic Fe₃O₄@COOH and Fe₃O₄@OH.

Line73-89.

The surface modification of small organic molecules and high polymer can inhibit the aggregation of Fe₃O₄ and increase its surface effect. Previous studies that the removal efficiency of pollutants in wastewater was significantly improved by introducing active groups on the surface of magnetic Fe₃O₄ nanoparticles modified by organic small molecules and organic macromolecules.[36-38] In this study, small organic molecules (sodium citrate, oleic acid, APTES) and organic polymers (D-dextran) were used to modify the magnetic nanoparticles. Sodium citrate and oleic acid can provide a large amount of -COOH, which can change the stability of magnetic Fe₃O₄ nanoparticles in water. Among them, the particles modified by sodium citrate have better hydrophilicity and dispersion, while the particles modified by oleic acid have better lipophilicity and are easier to be affinity with oil pollution pollutants. -OH (from dextran) and -NH₂ (from APTES) have strong complexing ability, in which -OH can make the particles have stronger metal cation adsorption function, the surface modified particles with -NH₂ can combine with metal anions through electrostatic adsorption under acidic conditions. In this study, the above polymers containing functional groups (hydroxyl, carboxyl, amino) were used to coat the cell surface of SY095, and the effects of different active groups on cells were evaluated by investigating their degradation performance in oily sewage and heavy metal sewage.

- ✓ Line 136: "The cells were coated with modified Fe₃O₄ NPs" Experimental section is not written well enough It should be in a way that the coating process can be reproducible even for non-expert researcher.

Line159-169.

The cells were coated with modified Fe₃O₄ NPs. We used ultraviolet spectrophotometer (Eppendorf, Germany) for the measurement of the optical density of the bacterial cells at 600 nm (OD₆₀₀) to monitor their growth rate. The optimum weight ratio for cell immobilization : DCW: modified Fe₃O₄ NPs was 1:2. According Fig.5a, calculate the dry cell weight (DCW) from the OD value reduction. When the bacterial cells were cultured for 36h, DCW of bacterial cell was 22.4g/L , the optical density of the bacterial cells at 600 nm (OD₆₀₀) was 0.72. The bacterial cells (from 100ml) were resuspended in a four-fold volume of mineral medium, and then the magnetic nanoparticles (4.48g) are then added to the cell (DCW2.24g) suspension and oscillated for 6 hours. After that, the cells coated with magnetic nanoparticles were separated from the uncoated cells by an external magnetic field. The collected magnetic cells were resuspended in a PBS buffer solution and stores at 4°C for a short time.

- ✓ In the XRD patterns, the authors should identify each the diffraction peak with miller

index or phases.

Fig.2a Wide-angle XRD patterns of Fe₃O₄ NPs and modified Fe₃O₄ NPs,

(a) Fe₃O₄, (b) Fe₃O₄@COOH NPs (CA-coated), (c) Fe₃O₄@COOH NPs (OA-coated), (d) Fe₃O₄@OH NPs, (e) Fe₃O₄@NH₂ (APTES-coated) NPs.

- ✓ In the hysteresis loops, it is difficult to distinguish the loop for each sample, the authors should modify Fig 2(b).

Fig.2b Magnetization curves of Fe₃O₄ NPs and modified Fe₃O₄ NPs

- ✓ The explanation of Ms modification of MNPs are insufficient. The authors should explain more detail, why Ms change.

Line 247-255.

Ms increased during OA modification, but there was no significant change during critical acid modification. The saturation magnetization of magnetic nanoparticles modified by oleic acid increased slightly, which was due to the decrease of crystallinity of magnetic nanoparticles modified by oleic acid. In addition, the surface modification can change the particle size of magnetic nanoparticles, which may be the reason for the increase of saturation magnetization of magnetic nanoparticles after oleic acid modification.

Appendix B

Dear Editor,

We are thankful for sending our manuscript for peer review. We are also thankful to the reviewers for their useful insights into our manuscript. We have considered all the comments and have acted accordingly. The changes made in the revised manuscript are highlighted in RED. The responses to the comments are given below.

Response to Reviewer: 1 comment

General comment

The topic is interesting and within the scope of the journal. After major revision, it may be suitable for publication.

Response

We are highly thankful to the reviewer for carefully evaluating our manuscript and his/her significant comments that improve the quality of the manuscript. We have gone through the points raised by the reviewer and address almost all of them. Below we provide a point-by-point response to each comment and make changes in the manuscript accordingly. The new changes in the manuscript are highlighted in red color.

Comment 1

Abstract has not started with appropriate sentences and it is recommended that the basic sentences of Abstract be improved.

Response:

We are thankful to the reviewer for his/her valuable suggestions. We have revised the abstract, start it with a proper sentence and correct the structure of all the sentences in the abstract.

Comment 2

There are a lot of grammatical, typographical, and punctuation errors in this manuscript that should be carefully checked and corrected.

Response:

We have thoroughly revised our manuscript with an expert native speaker editor and corrected all grammatical, typographical, and punctuation errors.

Comment 3

There is no enough background given to understand the rationale. The Introduction part is not very well-structured and lacks a clear backbone.

Response:

We added more background information in the introduction section to make it easier for the readers to understand the rationale.

Comment 4

The figure 3 (SEM) is not clear and need to be replaced by figures with higher resolution.

Response

The image of figure 3 (1.96MB) was replaced with a higher-resolution version (8.68MB).

Comment 5

For table 2 and 3 , the conditions should be explain completely. Explain more and in detail.

Response:

We have provided a detailed explanation for Tables 2 and 3. Both were carried out by GC/MS. However, the incubation time was different for each. In the Table caption, we mention the conditions for evaluating the properties of degradation products of cetane determined and crude oil degradation products determined.

Comment 6

The Fig 1 (FTIR) is important and has not been described completely. Explain it more.

Response:

The reviewer's suggestion is well taken. We have explained the Fig 1 (FTIR) in detail in the revised version of our manuscript.

Comment 7

Table 1, the numbers should be rounded to two decimal places.

Response:

We have rounded the numbers to two decimal places.

Comment 8

Complete and acceptable explanations and interpretations should be provided for fig 7

Response:

We have added an explanation for Fig 7.

Comment 9

The conclusion section should be improved. Conclusions should be supported by the data.

Response:

We have improved the conclusion section in the revised draft of our manuscript.

Comment 10

In the introductory section, the author is suggested to refer the cited articles. At the end of the text:“ However, the synthesis of MNPs with consistent shapes and proven dispersion remains a challenge.”

It is suggested to refer to the following article:

[1] R. Jalilian and A. Taheri, Synthesis and application of a novel core-shell-shell magnetic ion imprinted polymer as a selective adsorbent of trace amounts of silver ions. *e-Polymers*, 18 (2018) 123-134. DOI: 10.1515/epoly-2017-0108

[2] A. Talavari, B. Ghanavati, A. Azimi and S. Sayyahi, PVDF/ MWCNT hollow fiber mixed matrix membranes for gas absorption by Al₂O₃ nanofluid *Progress in Chemical and Biochemical Research*, 4 (2021) 177-190. DOI: 10.22034/pcbr.2021.270178.1177

[3] M. Shahamatpour, S.M. Tabatabaee Ghomsheh, S. Maghsoudi and S. Azizi, Fenton Processes, Adsorption and Nano Filtration in Industrial Wastewater Treatment. *Progress in Chemical and Biochemical Research*, 4 (2021) 31-43. DOI: 10.22034/pcbr.2021.118152

[4] Z. Rezayati zad, B. Moosavi and A. Taheri, Synthesis of monodisperse magnetic hydroxyapatite/Fe₃O₄ nanospheres for removal of Brilliant Green (BG) and Coomassie Brilliant Blue (CBB) in the single and binary systems. *Advanced Journal of Chemistry-Section B*, 2 (2020) 159-171. DOI: 10.33945/sami/ajcb.2020.3.8

[5] H. Thacker, V. Ram and P.N. Dave, Plant mediated synthesis of Iron nanoparticles and their Applications: A Review. *Progress in Chemical and Biochemical*

Research, 2 (2019) 84-91. DOI: DOI: 10.33945/SAMI/PCBR.183239.1033

Response:

The reviewer's suggestion is well taken, and we have cited all the above mention studies in the introduction section of our revised manuscript.

Comment 11

Please compare the performance of this method with some previously reported works.

Response:

We have added a paragraph in the revised manuscript comparing the previous method and the current approach.

Comment 12

From my perspective of view, the present work is far from satisfaction. I would suggest the authors to include the drawbacks and possible future work for improvement in the conclusion part.

Response:

We have added limitations of the current methods at the end of the manuscript. Moreover, we also added a future perspective for improving the current methods and approaches for designing crude oil degradation.

Comment 12

Add some references related to the 2021 or 2020 research topic.

Response:

We have added updated references from 2021 and 2020 in the revised draft of our manuscript.

Response to Reviewer: 2 comments

Comment 1

Ms of Fe₃O₄ was 38.2. Basically, Ms of Fe₃O₄ coated by non-magnetic material will decrease. However, the authors reported Ms of Fe₃O₄@COOH (OA-coated) slightly increased. Why?

Response:

There may be various reasons. One possible reason for the different conditions, as mention in our study we optimize the condition according to experimental needs.

Which may slightly increase it. Moreover, another possible reason is the addition of carboxylic acid, which may also slightly increase the ms.